# Spin–Orbit and Zeeman Effects on the Electronic Properties of Single Quantum Rings: Applied Magnetic Field and Topological Defects

**DOI:** 10.3390/nano13091461

**Published:** 2023-04-25

**Authors:** José C. León-González, Rafael G. Toscano-Negrette, A. L. Morales, J. A. Vinasco, M. B. Yücel, H. Sari, E. Kasapoglu, S. Sakiroglu, M. E. Mora-Ramos, R. L. Restrepo, C. A. Duque

**Affiliations:** 1Grupo de Materia Condensada-UdeA, Instituto de Física, Facultad de Ciencias Exactas y Naturales, Universidad de Antioquia UdeA, Calle 70 No. 52-21, Medellín 050010, Colombia; jose.leong@udea.edu.co (J.C.L.-G.); alvaro.morales@udea.edu.co (A.L.M.); juan.vinascos@udea.edu.co (J.A.V.); 2Departamento de Física y Electrónica, Universidad de Córdoba, Carrera 6 No. 77-305, Montería 230002, Colombia; 3Department of Physics, Faculty of Science, Akdeniz University, 07058 Antalya, Turkey; myucel@akdeniz.edu.tr; 4Department of Mathematical and Natural Science, Faculty of Education, Sivas Cumhuriyet University, 58140 Sivas, Turkey; sari@cumhuriyet.edu.tr; 5Department of Physics, Faculty of Science, Sivas Cumhuriyet University, 58140 Sivas, Turkey; ekasap@cumhuriyet.edu.tr; 6Dokuz Eylul University, Faculty of Science, Physics Department, 35390 Izmir, Turkey; serpil.sakiroglu@deu.edu.tr; 7Centro de Investigación en Ciencias, Instituto de Investigación en Ciencias Básicas y Aplicadas, Universidad Autónoma del Estado de Morelos, Av. Universidad 1001, Cuernavaca CP 62209, Morelos, Mexico; memora@uaem.mx; 8Universidad EIA, Envigado 055428, Colombia; ricardo.restrepo@eia.edu.co

**Keywords:** spin–orbit interaction, Zeeman splitting, circularly polarized light, topological defect, optical absorption coefficient

## Abstract

Within the framework of effective mass theory, we investigate the effects of spin–orbit interaction (SOI) and Zeeman splitting on the electronic properties of an electron confined in GaAs single quantum rings. Energies and envelope wavefunctions in the system are determined by solving the Schrödinger equation via the finite element method. First, we consider an inversely quadratic model potential to describe electron confining profiles in a single quantum ring. The study also analyzes the influence of applied electric and magnetic fields. Solutions for eigenstates are then used to evaluate the linear inter-state light absorption coefficient through the corresponding resonant transition energies and electric dipole matrix moment elements, assuming circular polarization for the incident radiation. Results show that both SOI effects and Zeeman splitting reduce the absorption intensity for the considered transitions compared to the case when these interactions are absent. In addition, the magnitude and position of the resonant peaks have non-monotonic behavior with external magnetic fields. Secondly, we investigate the electronic and optical properties of the electron confined in the quantum ring with a topological defect in the structure; the results show that the crossings in the energy curves as a function of the magnetic field are eliminated, and, therefore, an improvement in transition energies occurs. In addition, the dipole matrix moments present a non-oscillatory behavior compared to the case when a topological defect is not considered.

## 1. Introduction

The study of low-dimensional semiconductor heterostructures has attracted much attention from the scientific community due to their novel and prospective applications. Varying some of their compositional or configurational features—such as size and geometry—or applying external fields to them, their physical properties can be conveniently tuned. This allows for their use in various research fields such as electronics, optoelectronics, photonics, and sensing (see [1,2,3,4,5]). Within this realm, semiconductor quantum rings (QRs) are low-dimensional structures in which, according to a suitable design, the charge carriers can be confined to move in two dimensions. In these systems, the spectra are adjustable through various mechanisms, which allows from a theoretical and experimental point of view to search for the separations between energy levels that are suitable for the applications sought [6,7,8].

Among the possibilities that have been found to control the optoelectronic properties are shape and size; in the work of Ref. [9], there is a comparison to determine which of these two parameters generates the greatest changes, concluding that for the geometries used there, the size of the structures has a greater influence. Other examples in which the influence of geometry on electron confinement is demonstrated are in [10], where it is concluded that volcano shape potential demonstrates a high optical gain in InAsN/GaAs nanostructures. Ref. [11] studies the effects of the size of QRs on the lowest states. Several works have been dedicated to modeling the environment in which electrons and holes are confined; for this purpose, potentials defined by different functions have been used, including the simple square potential, the Tietz potential, which is used, for example by Khordad et al. [12], where it is considered a good manner to describe molecular dynamics for medium and high rotational-vibrational quantum numbers. The Morse-like potential was described for the first time in the article [13] to represent the movements of the nucleus in a diatomic molecule. Additionally, Nautiyal et al. [14] used the Kratzer potential in quantum dots with the inclusion of the spin–orbit effect, whose functional form tries to describe intermolecular vibrations.

Including external effects such as magnetic and electric fields has evidenced the loss of degeneracies, which improves the possibilities of finding behaviors different from those exhibited only by confinement. Such is the case in [15,16], where the breaking degeneration is observed because of magnetic field influence. In [17], the effects on direct and indirect exciton are studied in vertically coupled QRs under the application of an electric field, generating a triangular shape potential. Another work, Ref. [18], involves crater-like shaped QRs with an electric field generating a strong change in the dipole moment and polarizability.

The temperature and/or hydrostatic pressure are physical parameters that change the effective confinement of electrons or holes. In Ref. [19], the authors analyzed the effects of hydrostatic pressure and temperature, observing that with the increase in temperature, the second harmonic generation coefficient increases and blueshifts. Bala et al. [20] describe the effects on the absorption coefficient in Ga${}_{0.7}$In${}_{0.3}$N/GaN QRs due to pressure and temperature. Irregularities in materials are found because of impurities’ presence; they become zones of attraction or repulsion and create Coulomb-type interactions due to the impurity and carrier charges. The influence of a shallow donor impurity on the excitonic contribution in GaAs/AlGaAs and CdTe/CdSe truncated cone quantum dots can be found in [21]. Ghanbari et al. [22] studied the effects of impurity presence on thermodynamic parameters (heat capacity, free energy, entropy, mean energy, and Gibbs free energy).

When spin–orbit interaction (SOI) coupling is considered, several new states appear in the electronic structure concerning the electron spin up and spin down. An externally applied magnetic field implies the coupling with the electron spin; the associated interaction is called the Zeeman effect. The SOI effect can be significant under certain circumstances and is present without a magnetic field.

In low-dimensional semiconductor systems, the SOI may contribute through two distinct mechanisms. The first one is the Rashba SOI coupling [23,24], which arises from the asymmetry of the structure and describes the spin–orbit interaction in low-dimensional systems. In the particular case of heterostructures (semiconductor materials and interfaces between materials), it depends on the shape of the confinement potential. The second one is the Dresselhaus coupling [25], which arises from the lattice inversion asymmetry. In particular, the Dresselhaus coupling describes the spin–orbit interaction caused by the material’s crystal structure asymmetry. It is significant for semiconductor materials, where the spin–orbit interaction is weaker than in metals. The Dresselhaus Hamiltonian can be written as a function of the orbital angular momentum and spin operators and has the form of a square matrix. It is a function of the direction of the external magnetic field and the material’s properties, such as the effective mass of the electrons and the lattice constant. The SOI may be important for understanding electrical conductivity in carbon nanotubes and quantum dots, where the SOI may cause anisotropy in conductivity, which may have implications for creating electronic devices. Similarly, the SOI also contributes to understanding the physics of spin systems, such as topological spin systems and spin qubits in quantum computing. The SOI can be used to control spin orientation, which is essential for creating spin-robust qubits. All this finds its application in the design of new types of spin-based devices for spintronics. For multiple applications of the SOI, see, for example, Shakouri et al. [26] and references therein.

Several studies have concerned the SOI effects on quantum dots and QRs. For example, Splettestoesser et al. studied the spin–orbit coupling in persistent currents and identified that the spin current is not proportional to the charge current [27]. Castaño-Yepes and collaborators investigated the impact of a topological defect and the SOI on the thermomagnetic and optical properties in a two-dimensional quantum dot [28]. Pietiläinen and Chakraborty investigated the SOI effects on the electronic properties of a few interacting electrons in parabolic quantum dots finding that such interaction profoundly influences the electron energy spectrum [29]. A study on the SOI and Coulomb effects on the electronic charge and spin density distributions in InAs- and InSb-based QRs was performed in Ref. [30]. The influence of temperature on the nonlinear optical responses of a GaAs QR under simultaneous Dresselhaus and Rashba SOI contributions and Zeeman effect was considered in [31,32]. In the work of Ref. [33], a two-dimensional (2D) finite element method (FEM) is used to solve the electronic states of elliptical GaAs/AlGaAs QR. Effects of external fields, SOI, and eccentricity are explicitly considered, and the linear optical absorption coefficient is reported.

The topic of SOI effects has many open questions that merit further investigation. To a large extent, the investigations reported in the available literature have focused on systems with abrupt potential barriers and quasi-two-dimensional heterostructures with constant heights. In this sense, the effects of the height of the heterostructures have been fixed as a parameter that modifies the coupling constant of the Dresselhaus term [26]. In large part of QR research, the authors have considered external magnetic fields applied perpendicular to the structure, which preserves the axial symmetry in circular rings. In this way, the obtained wave functions are used efficiently in diagonalization problems of the n×n Hamiltonians, including SOI effects. The presence of in-plane electric fields breaks the axial symmetry of ring systems. The same situation is obtained by introducing topological defects in the geometry of the structures. The direct consequence of these last two effects translate into the enrichment of the optical transitions allowed between, for example, the system’s ground state and the first excited states. Possible applications of such effects include resonant radiation polarization detectors that can be tuned or adjusted through external electric fields. Finally, semiconductor heterostructures present interdiffusion problems at the interfaces of the materials that compose them. So, abrupt potential barriers are still a strong approximation in modeling physical systems.

Considering the open questions raised in the previous paragraph, we focus our research on the study of the effects of an external electric field on the optical and electronic properties of electrons confined in two-dimensional QRs. We analyze a 2D GaAs-based single QR whose confining potential model is described via an inverse quadratic function called the Hellmann potential and which allows efficient modeling of interdiffusion effects in the ring material. We also include the effects of a topological defect and report the linear optical absorption coefficient features related to intraband transitions in the presence of a magnetic field applied perpendicular to the structure. The optical properties are analyzed for interlevel transitions from the ground to the first two excited states. In the case of incident resonant radiation, we will study the effects of left- and right-hand circular polarization, which are combinations of mutually perpendicular linear polarizations. Calculations are made in the effective mass and parabolic conduction band approximations. SOI effects are introduced through a 2×2 Hamiltonian diagonalized by FEM using the licensed Comsol-Multiphysics 5.4 software. To implement the FEM calculations, we have used meshes with a spatial refinement that allows accounting for the divergences at the system’s center, typical of the confinement potential we have chosen. In our opinion, the study is novel, and we consider that it significantly contributes to the knowledge of magnetic properties in two-dimensional order systems considering SOI effects. The technique for the diagonalization of the n×n Hamiltonians and the type of problem without symmetries that we present here can be extended to the study of QRs of variable height by implementing the adiabatic approximation. In the same way, this research can be extended to the study of the effects of non-resonant high-intensity incident radiation with linear or circular polarization.

The main question of this article and that we answer through this research is: how an in-plane applied electric field and a topological defect can modify the magneto-optical properties relative to states of an electron confined in a quantum ring with Hellmann-like potential using circularly polarized resonant incident radiation.

This work is presented as follows: Section 2 shows the theoretical model, Section 3 is dedicated to results and discussion, and Section 4 presents the most outstanding results through conclusions.

## 2. Theoretical Model

It is considered an electron confined in a GaAs QR, under an inversely quadratic potential (Hellmann potential) in the presence of an electric field applied (in the +x direction with magnitude *F*), an applied magnetic field (in the +z direction with magnitude *B*), and SOI and Zeeman effects. Under the effective mass approximation, the Hamiltonian can be written as
(1)$$\hat{H}=\left[\frac{{\left(\hat{P}+q\hat{A}\right)}^{2}}{2\;m^{*}}+V\left(x,y\right)+q\;F\;x\right]I+\frac{1}{2}\;g\;B\;\mu_{B}\;\sigma_{z}+{\hat{H}}_{R}+{\hat{H}}_{D},$$
where $m^{*}$ is the electron effective mass, *q* is the absolute value of the electron charge, $\vec{A}=\frac{B}{2}\left(-y,x,0\right)$ is the magnetic vector potential, and V(x,y) is the 2D-confining potential. Moreover, *g* is the Landé factor, $\mu_{B}$ is the Bohr magneton, and *I* represents the 2×2 identity matrix. In Equation (Equation 1), ${\hat{H}}_{R}$ and ${\hat{H}}_{D}$ are the Rashba and Dresselhaus terms, respectively, which are given by [33]
(2)$${\hat{H}}_{R}=\frac{\alpha }{\hbar }\left(p_{y}\sigma_{x}-p_{x}\sigma_{y}\right)\;,$$
and
(3)$${\hat{H}}_{D}=\frac{\beta }{\hbar }\left(p_{x}\sigma_{x}-p_{y}\sigma_{y}\right),$$
where α and β are the coupling constants and $\sigma_{i}\;\left(i=x,y,z\right)$ are Pauli matrices.

The α constant measures the intensity of the spin–orbit interaction in a material which depends on the material’s properties and can vary widely between different materials and crystal structures. The β constant describes the spin–orbit interaction caused by the lack of symmetry in the material’s crystalline structure. The α and β constants can be experimentally determined by different techniques, such as photoemission spectroscopy, scanning tunneling spectroscopy, and the measurement of magnetic anisotropy in low-dimensional systems. Remarkably, the α constant is stronger in materials with an asymmetric crystal structure and low symmetry, such as materials with surfaces and interfaces. The determination of the Rashba and Dresselhaus coupling constants using the conductance of a ballistic nanowire has been reported by M. R. Sakr [34]. The determination of the Rashba spin–orbit coupling strength in InSb nanowire quantum dots under the influence of temperature and nuclear environment has been reported by M. Milivojevic [35]. The study was performed by measuring the magnetic susceptibility of the two-electron system in a double quantum dot. Meier et al. reported the measurement of Rashba and Dresselhaus spin–orbit magnetic fields [36]. They apply their method to GaAs/InGaAs quantum-well electrons, but it should be universally useful to characterize spin–orbit interactions in semiconductors and, therefore, could facilitate the design of spintronic devices.

In this work, the position-dependent 2D-Hellmann-like confinement potential reads
(4)$$V\left(x,y\right)=V_{0}\;\left[\frac{1}{2}-\frac{2\;R_{0}}{\sqrt{x^{2}+y^{2}}}+{\left(\frac{R_{0}}{\sqrt{x^{2}+y^{2}}}\right)}^{2}\right],$$
where $V_{0}$ is the potential barrier height and $R_{0}$ is the QR radius. The potential in Equation (Equation 4) is a combination of Coulomb and Yukawa potentials, where the minimum energy value is at $r=\sqrt{x^{2}+y^{2}}=R_{0}$ with a potential energy of $-V_{0}/2$. Figure 1a shows a graph of the Hellmann-like potential for $R_{0}=15$ nm and $V_{0}=228$ meV. Note that for r→∞, the confinement potential goes to the limit $V\to +V_{0}/2$.

Taking into consideration all the interactions described above and after some algebraic steps, the 2×2 matrix Hamiltonian can be written as
(5)$$\hat{H}=\left(\begin{array}{cc} H_{11} & H_{12}\\ H_{21} & H_{22}, \end{array}\right)\;,$$
where
(6)$${\hat{H}}_{0}=-\frac{\hbar^{2}}{2\;m^{*}}\left(\frac{\partial^{2}}{\partial x^{2}}+\frac{\partial^{2}}{\partial y^{2}}\right)+V\left(x,y\right)\;,$$
(7)$$H_{11}={\hat{H}}_{0}+\frac{q^{2}\;B^{2}}{8\;m^{*}}\left(x^{2}+y^{2}\right)+\frac{i\;q\;B\;\hbar }{2\;m^{*}}\left(y\frac{\partial }{\partial x}-x\frac{\partial }{\partial y}\right)+\frac{1}{2}g\;B\;\mu_{B}+q\;F\;x\;,$$
(8)$$H_{22}={\hat{H}}_{0}+\frac{q^{2}\;B^{2}}{8\;m^{*}}\left(x^{2}+y^{2}\right)+\frac{i\;q\;B\;\hbar }{2\;m^{*}}\left(y\frac{\partial }{\partial x}-x\frac{\partial }{\partial y}\right)-\frac{1}{2}g\;B\;\mu_{B}+q\;F\;x\;,$$
(9)$$H_{12}=\left(\alpha -i\;\beta \right)\frac{\partial }{\partial x}+\left(\beta -i\;\alpha \right)\frac{\partial }{\partial y}\;,$$
and
(10)$$H_{21}=-\left(\alpha +i\;\beta \right)\frac{\partial }{\partial x}-\left(\beta +i\;\alpha \right)\frac{\partial }{\partial y}\;.$$

Note that in Equations (7) and (8) we have used the fact that for the expansion of the kinetic energy term we are considering the Coulomb gauge ($\nabla \cdot \hat{A}=0$) together with the fact that ${\left(\hat{P}+q\;\hat{A}\right)}^{2}\;\psi ={\left(\hat{-i\;\hbar \nabla }+q\;\hat{A}\right)}^{2}\;\psi =-\hbar^{2}\nabla^{2}\;\psi +q^{2}\;A^{2}\;\psi -i\;\hbar \nabla \cdot \hat{A}\;\psi -i\;\hbar \;\hat{A}\cdot \nabla \;\psi$. Additionally, $\nabla \cdot \hat{A}\;\psi =\nabla \cdot \left(\hat{A}\;\psi \right)=\hat{A}\cdot \nabla \left(\psi \right)+\left(\nabla \cdot \hat{A}\right)\;\psi =\hat{A}\cdot \nabla \left(\psi \right)$. See, for instance, Duque et al. [37] and Dahiya et al. [38].

The Schrödinger equation for the envelope wavefunction in the presence of SOI and Zeeman effects can be written as
(11)$$\left(\begin{array}{cc} H_{11} & H_{12}\\ H_{21} & H_{22} \end{array}\right)\left(\begin{array}{c} \psi_{↑}\\ \psi_{↓} \end{array}\right)=E\left(\begin{array}{c} \psi_{↑}\\ \psi_{↓} \end{array}\right)\;,$$
where $\psi_{↑}$ and $\psi_{↓}$ are the wavefunctions spatial part corresponding to the eigenvalues of spin up and spin down, respectively.

The solution of the matrix differential equation in Equation (Equation 11) has been obtained through the FEM implemented in the licensed software COMSOL-Multiphysics [39,40,41]. In Figure 1b,c, we show the geometry and mesh for this work’s finite element method calculation, without and with a topological defect, respectively. The circle with $R_{1}$ radius is defined to establish the Dirichlet boundary conditions. In Figure 1c, the Dirichlet boundary condition was also considered along the two straight radial lines. In Figure 1c, we show the topological defect introduced in the problem, which consists of eliminating from the circle of radius $R_{1}$ a circular sector of angle $2\pi -\theta_{0}$. Note that joining the two radial lines gives rise to a three-dimensional flat conical structure whose physics can be analyzed using the two-dimensional flat figure shown in Figure 1c, where the defect is included. The details of the mesh used for the calculation with the FEM are the following. Both in the absence and the presence of a defect, a double refinement is used in the region $r<R_{0}$ concerning the fine mesh for $R_{0}<r<R_{1}$. This is to address the divergence of the potential at r→0. In both cases, with and without defects, a triangular element mesh is used. In the absence of a defect, the mesh parameters are 1554 mesh vertices, 3046 triangles, 156 edge elements, 8 vertex elements, 0.3202 for the quality element minimum, 0.8152 for the medium element quality, 0.01373 for the element area ratio, and 7840 nm${}^{2}$ for the mesh area. When the defect is considered, the corresponding parameters are 1322 mesh vertices, 2544 triangles, 178 edge elements, 11 vertex elements, 0.3259 for the quality element minimum, 0.8158 for the medium element quality, 0.01969 for the element area ratio, and 7623 nm${}^{2}$ for the mesh area.

To study the optical absorption coefficient associated with the lowest confined electron states in the QR, the expression resulting from an approach based on the density matrix formalism is used [42]. Optical transitions are considered between the ground state and the first two excited states, with interlevel energy transitions denoted $E_{12}$ and $E_{13}$, respectively. A circularly polarized resonant incident radiation is used to excite the optical transitions. So, the linear optical absorption coefficient is given by
(12)$$\alpha^{\left(1\right)}\left(\omega \right)=\sqrt{\frac{\mu_{0}}{\varepsilon_{r}\;\varepsilon_{0}}}\;\;\sum_{j=2}^{3}\frac{\omega \;e^{2}\;\sigma \;\hbar \;\gamma_{1j}\;{\left|M_{1j}\right|}^{2}}{{\left(E_{1j}-\hbar \;\omega \right)}^{2}+{\left(\hbar \;\gamma_{1j}\right)}^{2}},$$
where ω is the incident photon frequency, σ is the carrier density, and $|M_{1j}|$ is the dipole matrix moment (DMM), which is given by
(13)$$|M_{1j}|=\int \int \left(\psi_{j↑}^{*}\;\zeta \;\psi_{1↑}+\psi_{j↓}^{*}\;\zeta \;\psi_{1↓}\right)\;dx\;dy\;,$$
where ζ is the incident light polarization. This work considers both right circular polarization $\zeta =\frac{x+i\;y}{\sqrt{2}}$, and left circular polarization, $\zeta =\frac{x-i\;y}{\sqrt{2}}$.

In Equation (Equation 12), Γ is the phenomenological operator responsible for the damping due to the electron–phonon interaction, collisions among electrons, etc. In this work, we assume that Γ is a diagonal matrix and its element $\gamma_{1j}$ (j=2,3) is the inverse of the relaxation time for the state $\psi_{j}$ [43]).

## 3. Results and Discussion

The parameters used for the calculations are: $R_{0}=15$ nm, and $m^{*}=0.067\;m_{0}$ is the GaAs effective mass, where $m_{0}$ is the free electron mass, $V_{0}=228$ meV, $\mu_{B}=4\;\pi \times 10^{-7}$ H m, $\hbar \gamma_{12}=\hbar \gamma_{13}=0.5$ meV, and $\varepsilon_{r}=12.58$ [33]. For the α-, β-, and *g*-parameters we choose β=10.8 meV nm, α=β/2, g=−2.15, $\sigma =3.0\times 10^{22}$ m${}^{-3}$ [26]. The β=10.8 meV nm value is obtained from the bulk Dresselhaus constant ($\beta_{b}$) as $\beta =\pi^{2}\;\beta_{b}/d$, where d=5 nm is the typical height of the ring structure in the growth direction and $\beta_{b}=27.5$ meV nm${}^{3}$ for GaAs.

Figure 2 shows the first five energy levels of an electron confined in the QR as functions of the applied magnetic field, for electric field values of F=0 and F=0.5 kV/cm, without SOI and Zeeman effects. Calculations are without topological defects. At zero electric field, Figure 2a, the energies exhibit oscillatory behavior, where crossings in the energy levels are noted for some magnetic field values. For instance, at B=2.5 T a crossing is observed between the ground state and the first excited state. At this point, the ground state wavefunction changed symmetry; this phenomenon is known as Aharonov–Bohm oscillations. When the electric field is F=0.5 kV/cm, the ground state behaves differently, and crossings with the first excited state are eliminated. This happens because the electric field application locates the charge carrier closer to the potential hard core, and spatial symmetry breaks. In addition, applying the electric field generates anticrossings between the first and second excited states for two applied magnetic field values (circles marked in Figure 2b). Without the SOI and Zeeman effect, each energy level is doubly degenerate.

Figure 3 shows the QR’s first eight electron energy levels as a function of the applied magnetic field for electric field values of F=0 and F=0.5 kV/cm, without the SOI effect and with Zeeman splitting. In both cases, Figure 3a,b, it is noted that the ground state is twice degenerate for zero magnetic fields, and the first excited state is four times degenerate. However, the degeneration breaks when the magnetic field differs from zero, and each level splits into two. The splitting increases as the magnetic field intensifies, ratifying the Zeeman effect in the system. The inclusion of the Zeeman effect preserves the oscillations in the energy levels. Consequently, crossings between states occur at the same magnetic field values as those shown in Figure 2a; that is, the crossing between the ground state and the first excited state (arising from the splitting of the ground state shown in Figure 2) occurs at B=2.5 T but for a higher energy value as compared with Figure 2a. This means there is a slight increase in energy by including the Zeeman effect. When the applied electric field strength is raised to F=0.5 kV/cm, a similar behavior to that in Figure 2b is observed, where the crossings between the ground state and the first excited state are eliminated. In addition, the ground state is affected by the application of the electric field due to a more significant localization of the electron towards the potential hard core in the QR, and, therefore, the energy becomes lowered, compared to Figure 2a. In this case, it is also observed that the application of an electric field produces anticrossings between some states for some particular magnetic field values; a particular case is the anticrossing indicated with the circle in Figure 3.

Figure 4 shows the first six electron energy levels in the QR as functions of the applied magnetic field for electric field values of F=0 (a, c) and F=0.5 kV/cm (b, d). In (a, b)/(c, d), the calculations are without/with the topological defects. Results are considering the inclusion of both SOI and Zeeman splitting effects. When F=0, Figure 4a, the energy levels also have an oscillatory character, so there are crossings between them. However, unlike Figure 3, the first and second excited states do not cross for any value of the magnetic field. Furthermore, the energy difference between these states at zero magnetic fields and 15T is smaller than in Figure 3a. This is explained by the fact that by including the SOI effect, the terms $H_{12}$ and $H_{21}$ in Equation (Equation 5) contribute significantly to variations in energy levels. When the electric field turns on to F=0.5 kV/cm, crossings between the ground state and the first excited state disappear. The same behavior can be noticed for the energy differences of these states at zero magnetic field and at 15T. A $10^{\circ }$ topological defect is included on the surface of the geometry as shown in Figure 1c, so the angle spanned by the system geometry is $\theta_{0}=350^{\circ }$. The energy curves as a function of the applied magnetic field are those shown in Figure 4c,d. It is noted that a topological defect substantially changes the energy spectra; at B=0, the states are doubly degenerate, and for magnetic fields other than zero, the degeneracy breaks, so each level is divided in two, and this division increases as the magnetic field increases. Furthermore, at B=0, the energies of the ground state for Figure 4c,d are higher than Figure 4a,b; this is because the topological defect increases the confinement of the electron in the QR. The topological defect slightly improves the 1→2 and 1→3 energy transitions compared to Figure 4a,b.

Figure 5 shows the lowest four energy levels for a confined electron in a QR under the inversely quadratic potential as a function of the $\theta_{0}$-parameter. It should be noted that $\theta_{0}$ is the angular section that encompasses the system geometry; therefore, the topological defect to obtain the energy curves shown in Figure 5 is in the interval from $1^{\circ }$ to $60^{\circ }$. It is noted that there is a decrease in the energies with increasing $\theta_{0}$ since the states are less confined, and therefore, the energy levels are reduced. When $\theta_{0}=360^{\circ }$, the geometry of the system does not present a topological defect, therefore the energies obtained for the electric and magnetic fields of F=0.5 kV/cm and B=0.5 T are equal to those shown in Figure 4b when B=10 T (see red stars at the right-hand vertical axis). It is also important to note that for $\theta_{0}=360^{\circ }$, the energy values are below the trend that the energies have for the previous $\theta_{0}$-value. This is because the boundary conditions defined from the exterior $R_{1}$ of Figure 1c change as the $\theta_{0}$-parameter increases, so the wavefunction is not spatially uniformly distributed in the neighborhoods of $R_{0}$. When reaching $360^{\circ }$, the boundary conditions of the problem are returned to the original, as shown in the blue line in Figure 1b; consequently, the wavefunction recovers its uniform distribution.

In Figure 6, the squared off-diagonal DMMs corresponding to the investigated 1→2 and 1→3 transitions are shown as functions of the applied magnetic field, for fixed electric field strengths, with α=β=0 and *g* = −2.15. In Figure 6a,c, the DMMs exhibit a piecewise quasi-rectangular shape (slight linear-like dips appear as the field intensifies), including intervals of zero value, for example, 0–2.4 T, 3–7.5 T, and 9–12.7 T for the transition 1→2, between 0.1–2.9 T, 5–8.9 T, and 10–15 T for the transition 1→3 with the right circular polarization and F=0, in the same way, other intervals can be found with DMM equal to zero for the left circular polarization, as can be seen in Table 1. A general magnitude decrease is observed for these two quantities with the increasing magnetic fields outside the zero value intervals. The latter can be explained by the change in wavefunction symmetry associated with energy level crossings, as discussed above in the analysis of Figure 3a. When an electric field of F=0.5 kV/cm is applied, as in Figure 5b,d, the dependence of the DMMs on the intensity of the magnetic field maintains a well-defined shape, presenting an oscillatory behavior for magnetic fields greater than 7.2 T for the transition 1→2 and less than 11.1 T for the transition 1→3; note that when the right circular polarization presents maximum values, the left circular polarization has minimum values. Therefore, the oscillating pattern is observed due to applying an electric field. Additionally, the DMM values are reduced compared to when no electric field is applied. For both polarizations, there is no magnetic field value where $|M_{12}{|}^{2}$ and $|M_{13}{|}^{2}$ are simultaneously different from zero. By having two types of circular polarization, Figure 6a,c (or Figure 6b,d) are complementary, so when added together, you obtain a graph where the DMMs are approximately constant.

The situation in which the Hamiltonian in Equation (Equation 5) has non-zero off-diagonal terms associated with SOI brings about a different picture of the non-permanent electric polarization in the system. In Figure 7, the squares of 1→2 and 1→3 DMMs are shown as functions of the applied magnetic field for the same two values of electric field intensity previously considered. Calculation was performed using β=10.8 eV nm, α=β/2, and g=−2.15. For all cases in Figure 7, $M_{12}$ is zero for B=0, since the ground state and the first excited state are degenerate, and by the selection rules, this transition is forbidden. Additionally, for that same value of the magnetic field, $|M_{13}|\neq 0$ because the wavefunctions corresponding to these states have different parity, so $\int \int \Psi_{j}^{*}\left(x,y\right)\zeta \Psi_{i}\left(x,y\right)\;dx\;dy$ does not vanish. When the polarization is right- and left-circular, respectively, and F=0, the DMM of 1→2 transition oscillates as the applied magnetic field increases. In contrast, 1→3 transition does not have a monotonic behavior. When F=0.5 kV/cm, Figure 7b,d, an overall decrease in the magnitude of the DMM is noted, compared to the zero electric field case; this is explained by the fact that the electric field spatially localizes the wavefunction and, consequently, the DMM diminishes. It is observed that when taking SOI and Zeeman effects into account, the DMMs $|M_{12}{|}^{2}$ and $|M_{13}{|}^{2}$ are non-zero for a single applied magnetic field value, which implies that the linear absorption coefficient has the contribution of the 1→2 and 1→3 transitions. As it happened in Figure 6, it can be noticed that the pairs of Figure 7a,c with Figure 7b,d are complementary.

Figure 8 shows the squares of the DMMs as a function of the $\theta_{0}$-parameter for the 1→2 and 1→3 transitions of an electron confined in a QR under the inversely quadratic potential, with β=10.8 meV nm, α=β/2, and g=−2.15. It is noted that the contribution of the 1→2 transition is greater than that of the 1→3 transition for any value of $\theta_{0}$ in the right-circular polarization; a different behavior is observed for the left-circular polarization: for an interval of $\theta_{0}$ the contribution of the 1→2 transition is greater than 1→3. However, this situation changes for other intervals of $\theta_{0}$. This can be attributed to the change in the spatial distributions of the wavefunctions of each state as the angle varies; therefore, the overlap between them will change depending on the case being studied.

Table 1 and Table 2 show the DMMs $|M_{12}{|}^{2}$ and $|M_{13}{|}^{2}$ with the different polarizations and their corresponding transition energies $E_{12}$ and $E_{13}$ to study the linear optical absorption coefficients. To obtain the absorption coefficient curves, Equation (Equation 12) is used, where the two transitions 1→2 and 1→3 are considered. However, from Table 1, there is only a contribution from one of the two transitions; consequently, a peak appears in the absorption curve. In Table 2 it can be seen that most of the DMMs are different from zero; therefore, two peaks appear in the absorption curves, with different intensities.

In Figure 9, the contributions to the linear absorption coefficient coming from 1→2 and 1→3 electron transitions are shown as functions of the incident photon energy without the SOI effect, considering Zeeman splitting. All the figures were obtained at fields of B=0,4,8, and 12 T. By comparing the panels at the left-hand column with the corresponding at the right-hand column, it can be noted that the application of the electric field changes the position of the resonant peak to higher energies for all the magnetic fields considered; this is because the transition energies 1→2 and 1→3 increase when the electric field is applied. When the polarization is right-hand circular, as in Figure 9a,b, the absorption at B=0 and B=4 T occurs for the transition from 1→3 since the DMMs for this value of *B* are non-zero. For the other cases, B=8 T and B=12 T, the absorption peak corresponds to the 1→2 transition. It can be noted that the intensity in the absorption peaks can vary according to the type of polarization; in relation, it is worth taking into account that the linear absorption coefficient is proportional to the product $\omega \;|M_{ij}{|}^{2}$, which, according to our findings, presents an oscillating character due to the particular dependence of DMMs involved in the magnetic field for each case of polarization. In the left-hand circular polarization, Figure 9c,d, the predominant transition for B=0 and B=4 T, is 1→3, while for B=8 T and B=12 T is the 1→2 transition. In addition, the resonant energy displacement presents a non-monotonous behavior, since the transition energies, as a function of the magnetic field, vary in an oscillating manner.

In Figure 10, contributions to the linear absorption coefficient are plotted as functions of the incident photon energy, taking into account the SOI and Zeeman splitting effects. It was evident that with the two effects, there is a reduction or increase in the intensity of the absorption coefficient, compared with results without SOI presented in Figure 9. The reason behind this phenomenon is the associated decrease or increase with the electric field of the DMMs —already discussed— when the two effects are included. The resonant peaks move to higher energies when F=0.5 kV/cm, because, in this case, the presence of the field eliminates the crossings for the lower energy states, see Figure 4b. Consequently, oscillations at the transition energies are considerably reduced compared to the cases where there was no applied electric field on the system. At B=0, the absorption curves present a single peak corresponding to the 1→3 transition, since for this value of the magnetic field, the ground state and the first excited state have the same energy, therefore $E_{12}=0$, consequently the product $\omega \;|M_{ij}{|}^{2}$ at the resonant energy is zero.

Figure 11 shows the linear absorption coefficient as a function of the incident photon energy for the 1→2 and 1→3 transitions, considering the effects of SOI, Zeeman splitting, and two cases where there is a topological defect of $30^{\circ }$ and $60^{\circ }$ in the geometry of the structure. It is observed that when the $\theta_{0}$-parameter increases, the resonant peaks shift slightly to lower energies since there is a decrease in the $E_{12}$ and $E_{13}$ energy transition as $\theta_{0}$-increases, as it is evidenced in Figure 5. The resonant peaks with lower energy correspond to the 1→2 transition and those with higher energy to the 1→3 transition. The intensity of the resonant peaks is more significant in the left-hand circular polarization than in the right-hand since the DMMs for this type of polarization present a greater value.

## 4. Conclusions

By using the effective mass and parabolic conduction band approximations, we have theoretically investigated the electron energy states and the related linear optical absorption coefficient in a two-dimensional GaAs-based quantum ring with inverse parabolic confining potential under the effects of perpendicular/in-plane applied magnetic/electric field. Calculations include the perturbations on the electron spectra concerning a topological defect. The Rashba and Dresselhaus spin–orbit interactions and the Zeeman splitting are discussed. Our main findings can be summarized as follows: (i) the spin–orbit and Zeeman effects are small perturbations on the electronic energy spectrum when the magnetic field is turned on; (ii) with and without spin–orbit effects, the ground electron state shows the well-known Aharonov–Bohm oscillations when the axial symmetry of the system is preserved (these oscillations became quenched for non-zero values of the applied electric field and also when the topological defects are considered); (iii) the optical absorption coefficient is strongly dependent on the incident light polarization and the applied electric and magnetic field strengths; (iv) the resonant peak of the optical absorption coefficient decreases when the spin–orbit interaction and Zeeman effects are turned on; and finally, (v) the applied electric field and the consideration of topological defects are two factors that enrich the allowed optical transitions in quantum rings and make these systems usable as polarization detectors. We hope this study stimulates further investigations in which 3D quantum rings that present non-regular variations in their height along the growth direction of the heterostructure are considered.

## Figures and Tables

**Figure 1 nanomaterials-13-01461-f001:**
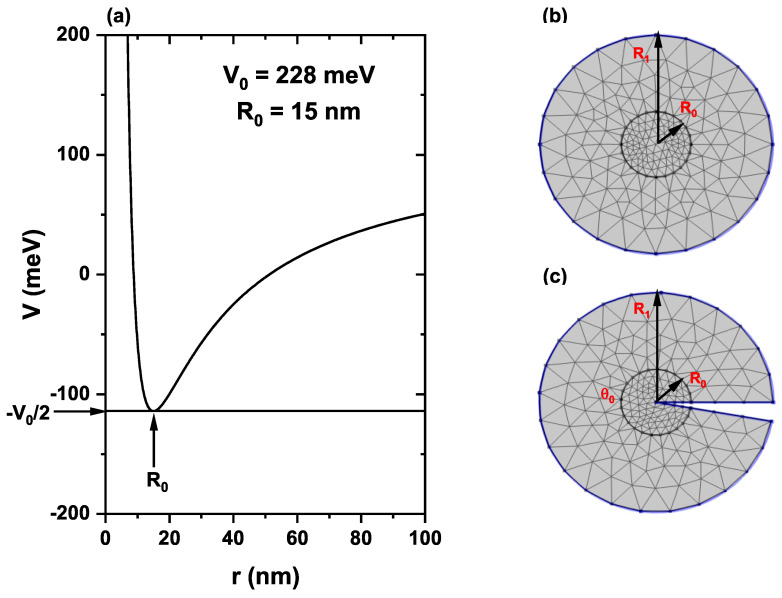
(**a**) Inversely quadratic potential (Hellmann-like potential) as a function of the *r*-radial coordinate for $R_{0}=15$ nm and $V_{0}=228$ meV. Figures (**b**,**c**) show the geometry and mesh for the finite element method calculation used in this work, without and with a topological defect (circular sector with $2\pi -\theta_{0}$ angle), respectively. The circle with $R_{1}$ radius is defined to establish the Dirichlet boundary conditions. In (**c**), the Dirichlet boundary condition was also considered along the two straight radial lines.

**Figure 2 nanomaterials-13-01461-f002:**
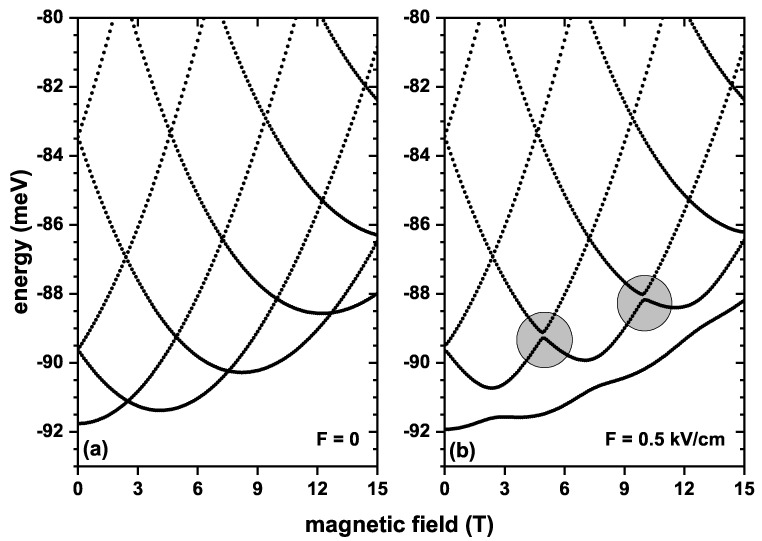
First five energy levels for a confined electron in a GaAs quantum ring under the inversely quadratic potential, as a function of the applied magnetic field, with F=0 (**a**) and F=0.5 kV/cm (**b**). Results are for α=β=0 and g=0.

**Figure 3 nanomaterials-13-01461-f003:**
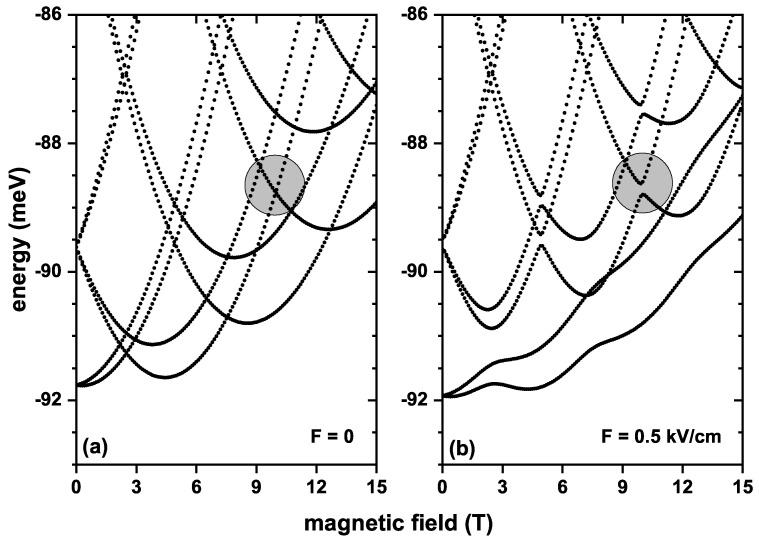
First eight energy levels for a confined electron in a GaAs quantum ring under the inversely quadratic potential, as a function of the applied magnetic field, with F=0 (**a**) and F=0.5 kV/cm (**b**). Results are for α=β=0 and g=−2.15.

**Figure 4 nanomaterials-13-01461-f004:**
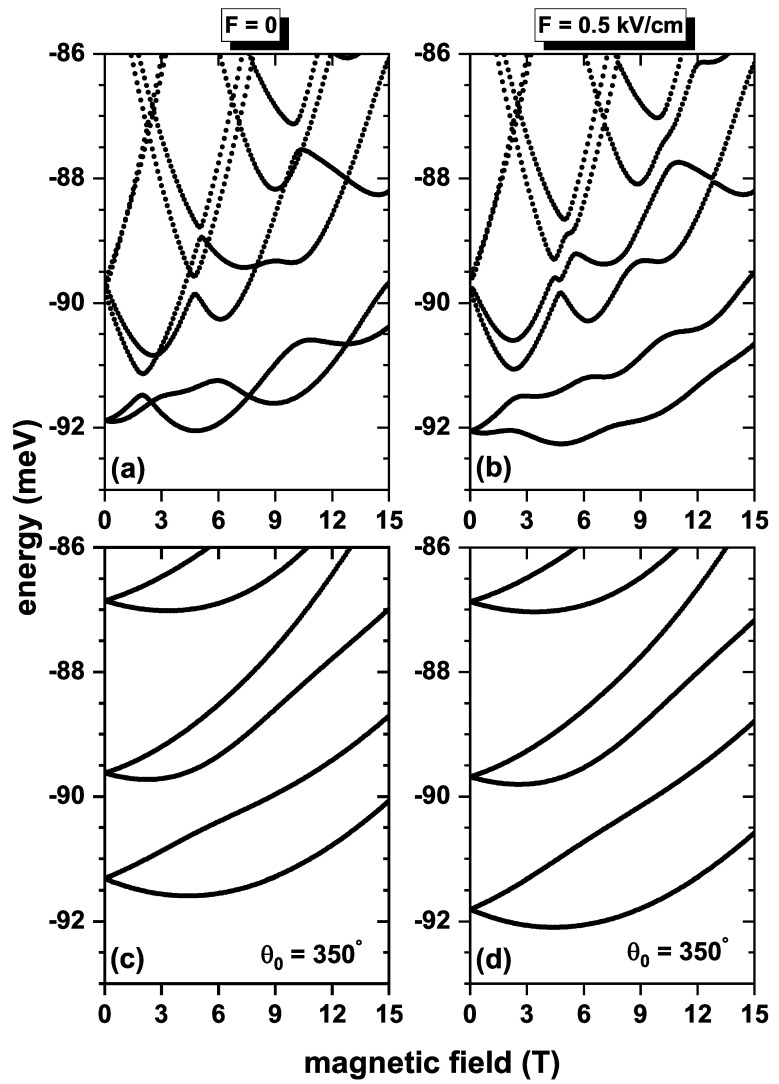
The lowest energy levels for a confined electron in a GaAs quantum ring under the inversely quadratic potential as a function of the applied magnetic field. Without topological defect (**a**,**b**) and with topological defect (**c**,**d**). Results are for β=10.8 meV nm, α=β/2, and g=−2.15.

**Figure 5 nanomaterials-13-01461-f005:**
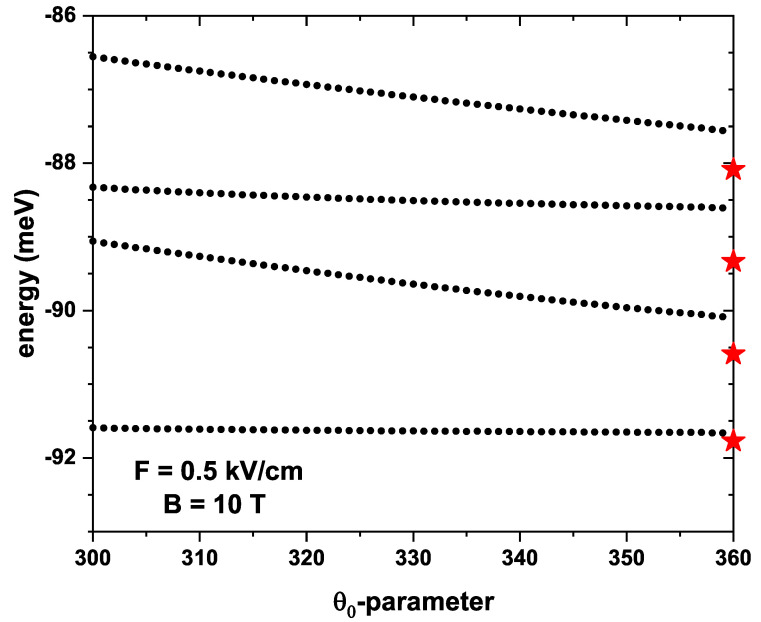
The lowest energy levels for a confined electron in a GaAs quantum ring under the inversely quadratic potential as a function of the angular $\theta_{0}$-parameter with F=0.5 kV/cm, B=10 T, β=10.8 meV nm, α=β/2, and g=−2.15. Stars at the right vertical axis are obtained without any topological defect limit.

**Figure 6 nanomaterials-13-01461-f006:**
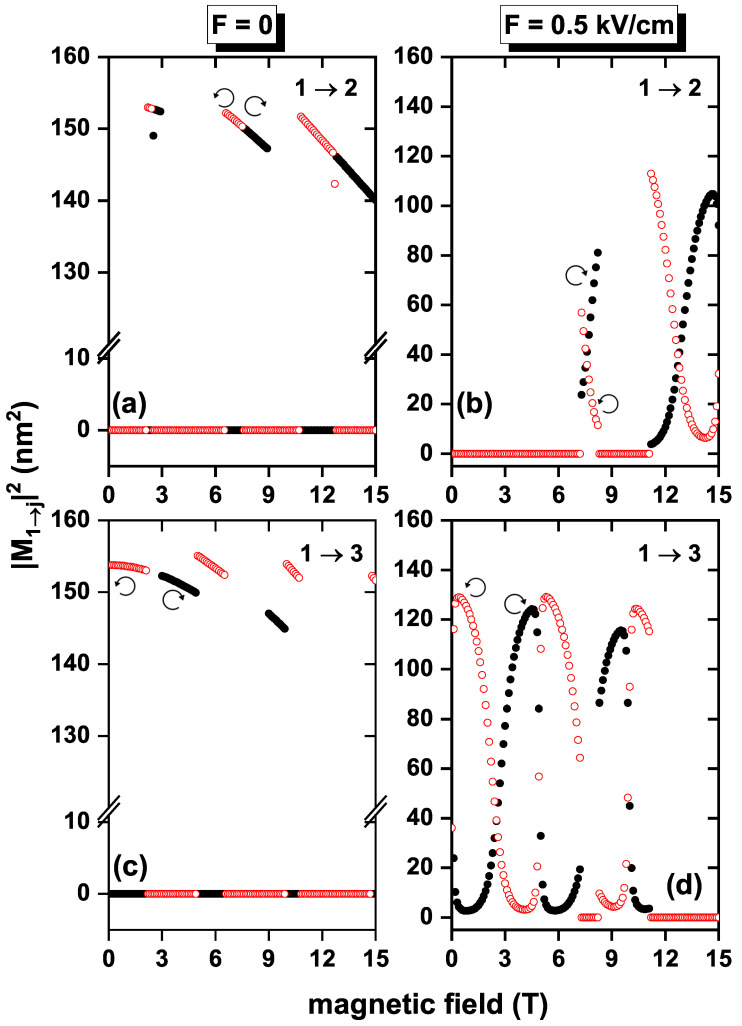
Squared dipole matrix moments as functions of the applied magnetic field for an electron confined in a GaAs quantum ring under the inversely quadratic potential for the transitions 1→2 (**a**,**b**) and 1→3 (**c**,**d**), with α=β=0 and g=−2.15. The empty circles (red color) correspond to left circular polarization and the full ones (black color) to right circular polarization. Figures (**a**,**c**) correspond to F=0 and (**b**,**d**) to F=0.5 kV/cm.

**Figure 7 nanomaterials-13-01461-f007:**
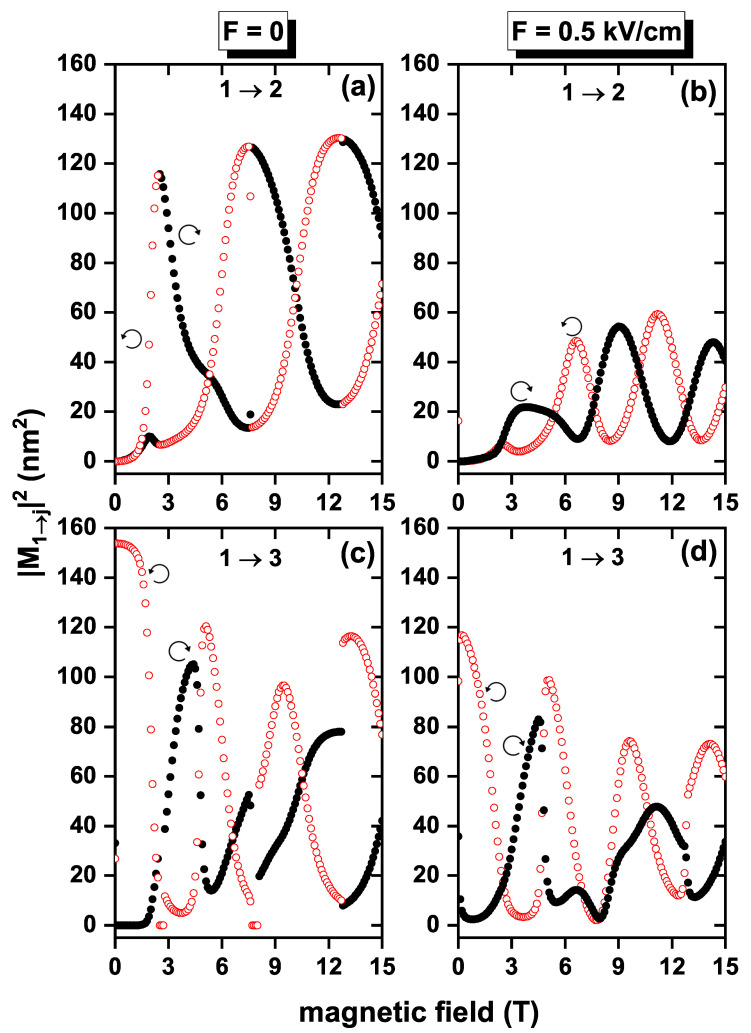
Squared dipole matrix moments as functions of the applied magnetic field for an electron confined in a GaAs quantum ring under the inversely quadratic potential for the transitions 1→2 (**a**,**b**) and 1→3 (**c**,**d**), with β=10.8 meV nm, α=β/2, and g=−2.15. The empty circles (red color) correspond to left-circular polarization, and the full ones (black color) to right-circular polarization. Figures (**a**,**c**) correspond to F=0 and (**b**,**d**) to F=0.5 kV/cm.

**Figure 8 nanomaterials-13-01461-f008:**
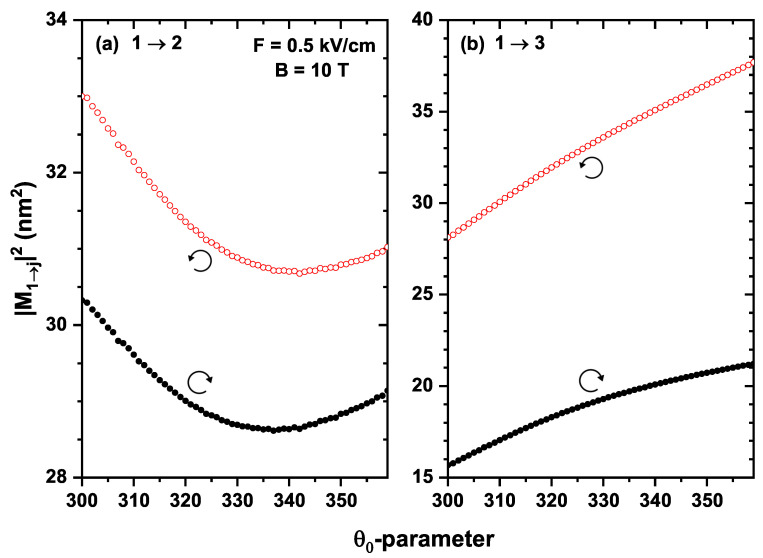
Squared dipole matrix moments as functions of the $\theta_{0}$-parameter for an electron confined in a GaAs quantum ring under the inversely quadratic potential, with F=0.5 kV/cm and B=10 T, for the transitions 1→2 (**a**) and 1→3 (**b**), with β=10.8 meV nm, α=β/2, and g=−2.15. The empty circles (red color) correspond to left-circular polarization, and the full ones (black color) to right-circular polarization.

**Figure 9 nanomaterials-13-01461-f009:**
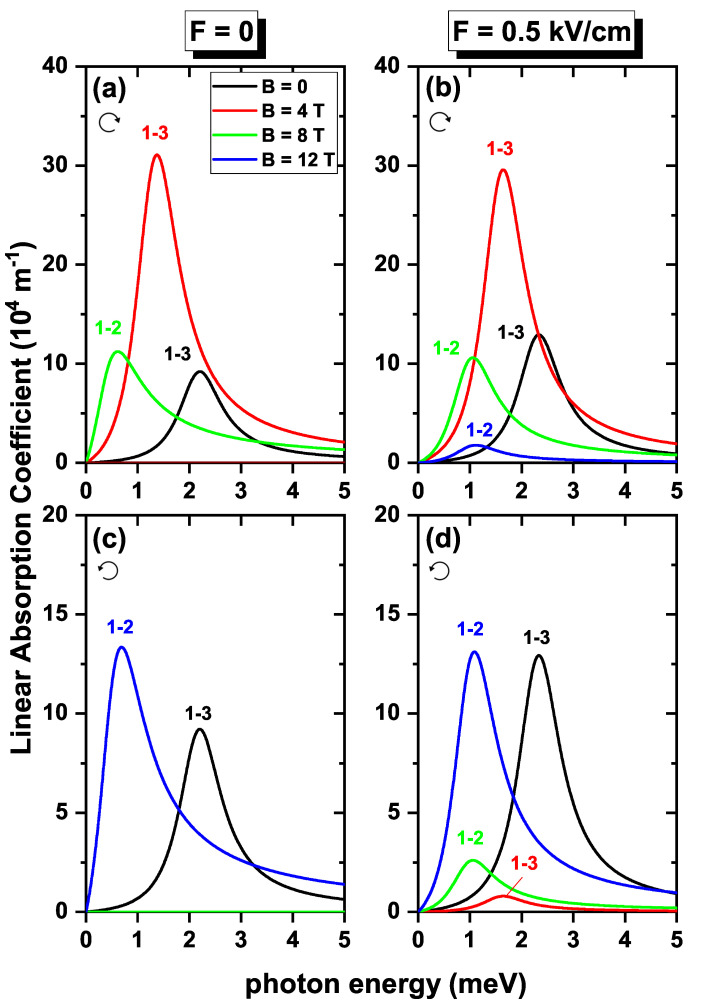
The linear optical absorption coefficient as a function of the incident photon energy for an electron confined in a GaAs quantum ring under the inversely quadratic potential, with α=β=0 and g=−2.15. Results are for F=0 (**a**,**c**) and F=0.5 kV/cm (**b**,**d**). The transitions considered correspond to right-hand circular (**a**,**b**) and left-hand circular (**c**,**d**) polarization for different values of applied magnetic fields.

**Figure 10 nanomaterials-13-01461-f010:**
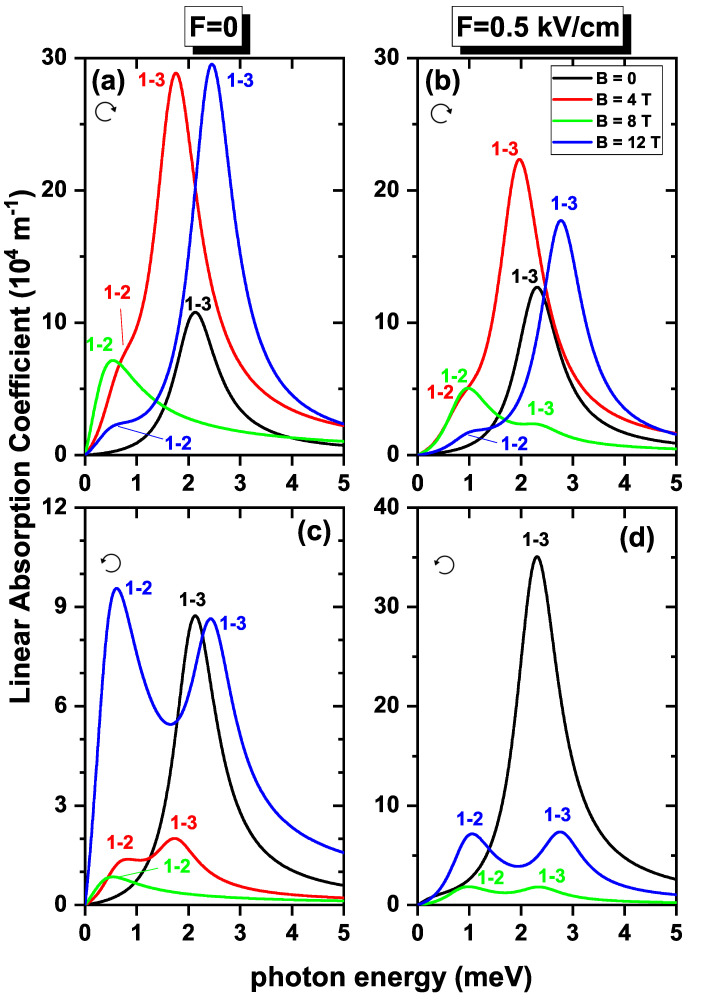
The linear optical absorption coefficient as a function of the incident photon energy for an electron confined in a GaAs quantum ring under the inversely quadratic potential, with β=10.8 meV nm, α=β/2, and g=−2.15. Results are for F=0 (**a**,**c**) and F=0.5 kV/cm (**b**,**d**). The transitions considered correspond to right-hand circular (**a**,**b**) and left-hand circular (**c**,**d**) polarization for different values of applied magnetic fields.

**Figure 11 nanomaterials-13-01461-f011:**
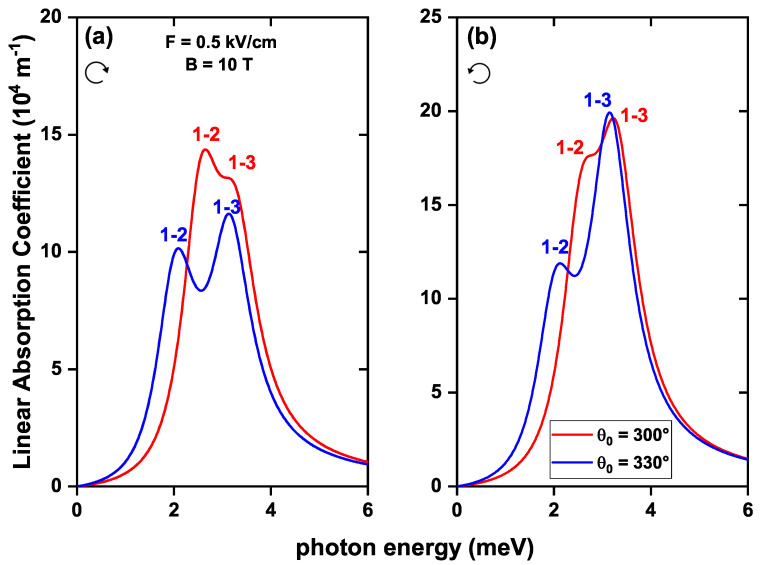
The linear optical absorption coefficient as a function of the incident photon energy for an electron confined in a GaAs quantum ring under the inversely quadratic potential, considering a topological defect in the structure’s geometry. Calculations are with right-hand (**a**) and left-hand (**b**) circular polarization, with β=10.8 meV nm, α=β/2, and g=−2.15.

**Table 1 nanomaterials-13-01461-t001:** The dipole matrix moments of the 1→2 and 1→3 transitions and their corresponding transition energies, for different values of electric and magnetic fields, with α=β=0 and g=−2.15.

α=0 β=0 g=−2.15	Right-Circular Polarization				F kV/cm	Left-Circular Polarization			
	B (T)					B (T)			
	0	4	8	12		0	4	8	12
$|M_{12}{|}^{2}$ (nm${}^{2}$)	0	0	149.2	2.1 × 10${}^{-5}$	0	0	0	3.2 × 10${}^{-6}$	148.4
$|M_{13}{|}^{2}$ (nm${}^{2}$)	27.3	151.2	0	0		27.3	3.2 × 10${}^{-6}$	0	0
$E_{12}$ (meV)	0	0.5	0.4	0.5		0	0.5	0.4	0.5
$E_{13}$ (meV)	2.1	1.1	1.0	1.5		2.1	1.1	1.0	1.5
$|M_{12}{|}^{2}$ (nm${}^{2}$)	0	0	68.8	10.9	0.5	0	0	16.8	82.3
$|M_{13}{|}^{2}$ (nm${}^{2}$)	36.1	118.9	0	0		36.1	3.9	0	0
$E_{12}$ (meV)	0	0.5	0.9	1.0		0	0.5	0.9	1.0
$E_{13}$ (meV)	2.3	1.6	1.0	1.5		2.3	1.6	1.0	1.5

**Table 2 nanomaterials-13-01461-t002:** The dipole matrix moments of the 1→2 and 1→3 transitions and their corresponding transition energies, for different values of electric and magnetic fields, with β=10.8 meV nm, α=β/2, and g=−2.15.

β=10.8 meV nm α=β/2 g=−2.15	Right-Circular Polarization				F kV/cm	Left-Circular Polarization			
	B (T)					B (T)			
	0	4	8	12		0	4	8	12
$|M_{12}{|}^{2}$ (nm${}^{2}$)	0	49.2	124.3	24.2	0	0	12.0	14.7	128.4
$|M_{13}{|}^{2}$ (nm${}^{2}$)	33.1	100.2	0	77.1		26.8	5.7	0	16.0
$E_{12}$ (meV)	0	0.6	0.2	0.3		0	0.6	0.2	0.3
$E_{13}$ (meV)	2.1	1.7	2.1	2.4		2.1	1.7	2.1	2.4
$|M_{12}{|}^{2}$ (nm${}^{2}$)	0	21.8	35.6	8.2	0.5	16.4	5.3	12.7	45.7
$|M_{13}{|}^{2}$ (nm${}^{2}$)	35.8	71	2.9	41		98.4	4.0	3.8	14.1
$E_{12}$ (meV)	0	0.7	0.8	0.9		0	0.7	0.8	0.9
$E_{13}$ (meV)	2.3	1.9	2.3	2.8		2.3	1.9	2.3	2.8

## Data Availability

No new data were created or analyzed in this study. Data sharing is not applicable to this article.

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
