# Peer review of "Spin–Orbit and Zeeman Effects on the Electronic Properties of Single Quantum Rings: Applied Magnetic Field and Topological Defects"

_nanomaterials, 2023, doi:10.3390/nano13091461_

Round 1

Reviewer 1 Report

In this theoretical work, the authors analyzed a 2D GaAs-based quantum ring with some inverse parabolic confining potentials. In particular, the authors looked at the electron energy states by numerical simulations with the COMSOL software. In their study, the authors do report some interesting findings related with the spin-orbit interactions as well as the Zeeman effect. They have performed numerical simulations to reveal the properties of electron energy states under various conditions. These theoretical observations may motivate further interest in the subject from the community.

After carefully going through the manuscript, I found the work is interesting and the presentation is well organized and written. A reader can access the materials without much technical difficulty. As the topic falls into the scope of the journal, and the quality of the manuscript meets the high standards, I would like to recommend it for publication in Nanomaterials.

If possible, it would be helpful to polish the English, as there are some grammar issues in the context.

Author Response

Referee 1

The Referee:

In this theoretical work, the authors analyzed a 2D GaAs-based quantum ring with some inverse parabolic confining potentials. In particular, the authors looked at the electron energy states by numerical simulations with the COMSOL software. In their study, the authors do report some interesting findings related with the spin-orbit interactions as well as the Zeeman effect. They have performed numerical simulations to reveal the properties of electron energy states under various conditions. These theoretical observations may motivate further interest in the subject from the community.

After carefully going through the manuscript, I found the work is interesting and the presentation is well organized and written. A reader can access the materials without much technical difficulty. As the topic falls into the scope of the journal, and the quality of the manuscript meets the high standards, I would like to recommend it for publication in Nanomaterials.

Our reply:

We would like to thanks to theReferee for his/her positive words about our work.

The Referee:

If possible, it would be helpful to polish the English, as there are some grammar issues in the context.

Our reply:

We have carefully checked the wording and grammar of our article. We believe that in its revised form it is free from many of the errors that have been typical of the way we traditionally express ourselves.

Reviewer 2 Report

In the presented work entitled „Spin-orbit and Zeeman effects on the electronic properties of single quantum rings: applied magnetic field and topological defects” by J. C. Leon-Gonzalez et al., the Authors investigate the effects of spin-orbit interaction (SOI) and Zeeman splitting on the electronic properties of an electron confined in a GaAs single quantum rings. They find that both SOI effects and Zeeman splitting reduce the absorption intensity for the considered transitions compared to the case when these interactions are absent. In addition, the magnitude and position of the resonant peaks have non-monotonic behavior with external magnetic fields. The most interesting finding in my opinion is that for the quantum ring with a topological defect the crossings in the energy curves as a function of the magnetic field are eliminated. Other minor findings are also reported.

The presented manuscript is written in a relatively clear and well organized manner, while the analysis seems to be free of any major errors. Unfortunately I cannot find much novelty in the presented results. In what follows I think that this paper will have little impact on the field and may be of low interest to the scientific community. In general: (i) the motivation for the conducted calculations is practically non-existent, (ii) the Authors conductor their calculations but under unknown hypothesis, (iii) the conclusions are simply a repetitive summary of the already described results, they do not provide the readers with any in-depth observations or findings that can be made based on the results.

I summary, the paper in its present form requires major improvements in terms of it presentation, aimed mainly at the motivation/hypothesis of conducted research but most importantly the Authors should answer the questions: what new findings and novel insights they can provided based on their results?

After such corrections I’m willing to reconsider this submission.

*end of report*

Author Response

Referee 2

The Referee:

In the presented work entitled “Spin-orbit and Zeeman effects on the electronic properties of single quantum rings: applied magnetic field and topological defects” by J. C. Leon-Gonzalez et al., the Authors investigate the effects of spin-orbit interaction (SOI) and Zeeman splitting on the electronic properties of an electron confined in a GaAs single quantum rings. They find that both SOI effects and Zeeman splitting reduce the absorption intensity for the considered transitions compared to the case when these interactions are absent. In addition, the magnitude and position of the resonant peaks have non-monotonic behavior with external magnetic fields. The most interesting finding in my opinion is that for the quantum ring with a topological defect the crossings in the energy curves as a function of the magnetic field are eliminated. Other minor findings are also reported.

Our reply:

We want to thank the Referee for his/her comments, which we consider extremely important. In our opinion, the Referee illustrates in a general way the scope of our investigation.

The Referee:

The presented manuscript is written in a relatively clear and well-organized manner, while the analysis seems to be free of any major errors. Unfortunately, I cannot find much novelty in the presented results. In what follows I think that this paper will have little impact on the field and may be of low interest to the scientific community. In general: (i) the motivation for the conducted calculations is practically non-existent, (ii) the Authors conductor their calculations but under unknown hypothesis, (iii) the conclusions are simply a repetitive summary of the already described results, they do not provide the readers with any in-depth observations or findings that can be made based on the results.

I summary, the paper in its present form requires major improvements in terms of its presentation, aimed mainly at the motivation/hypothesis of conducted research but most importantly the Authors should answer the questions: what new findings and novel insights they can provided based on their results?

After such corrections I’m willing to reconsider this submission.

Our reply:

We want to thank the Referee for his/her comments, which we consider extremely important.

We have presented in the Introduction section some of the details that have motivated this research. We have clearly and forcefully established the research question, which we have tried to answer in the broadest and most precise way throughout our article. Considering the breaking of the axial symmetry of the ring system given the presence of the in-plane electric field and the topological defect, we have presented in the sixth paragraph of the Introduction section the reason that justifies the inclusion in our study of the Rashba and Dresselhaus interaction effects on the Hamiltonian for the electron confined in the quantum ring under applied magnetic field effects.

Regarding the Rahba and Dreselhaus interactions we have added the following paragraph with its corresponding Reference:

In low-dimensional semiconductor systems, the SOI may contribute through two distinct mechanisms. The first one is the Rashba SOI coupling \cite{Vaseghi,Ali}, which arises from the asymmetry of the structure and describes the spin-orbit interaction in low-dimensional systems. In the particular case of heterostructures (semiconductor materials and interfaces between materials), it depends on the shape of the confinement potential. The second one is the Dresselhaus coupling \cite{Bejan}, which arises from the lattice inversion asymmetry. In particular, the Dresselhaus coupling describes the spin-orbit interaction caused by the material's crystal structure asymmetry. It is significant for semiconductor materials, where the spin-orbit interaction is weaker than in metals. The Dresselhaus Hamiltonian can be written as a function of the orbital angular momentum and spin operators and has the form of a square matrix. It is a function of the direction of the external magnetic field and the material's properties, such as the effective mass of the electrons and the lattice constant. The SOI may be important for understanding electrical conductivity in carbon nanotubes and quantum dots, where the SOI may cause anisotropy in conductivity, which may have implications for creating electronic devices. Similarly, the SOI also contributes to understanding the physics of spin systems, such as topological spin systems and spin qubits in quantum computing. The SOI can be used to control spin orientation, which is essential for creating spin-robust qubits. All this finds its application in the design of new types of spin-based devices for spintronics. For multiple applications of the SOI, see, for example, Shakouri \textit{et al.} \cite{Shakouri} and references therein.

Regarding the motivation for our article and the news that is introduced, we have written the following paragraphs:

The topic of SOI effects has many open questions that merit further investigation. To a large extent, the investigations reported in the available literature have focused on systems with abrupt potential barriers and quasi-two-dimensional heterostructures with constant heights. In this sense, the effects of the height of the heterostructures have been fixed as a parameter that modifies the coupling constant of the Dresselhaus term \cite{Shakouri}. In large part of QR research, the authors have considered external magnetic fields applied perpendicular to the structure, which preserves the axial symmetry in circular rings. In this way, the obtained wave functions are used efficiently in diagonalization problems of the $n\times n$ Hamiltonians, including SOI effects. The presence of in-plane electric fields breaks the axial symmetry of ring systems. The same situation is obtained by introducing topological defects in the geometry of the structures. The direct consequence of these last two effects translate into the enrichment of the optical transitions allowed between, for example, the system's ground state and the first excited states. Possible applications of such effects include resonant radiation polarization detectors that can be tuned or adjusted through external electric fields. Finally, semiconductor heterostructures present interdiffusion problems at the interfaces of the materials that compose them. So, abrupt potential barriers are still a strong approximation in modeling physical systems.

Considering the open questions raised in the previous paragraph, we focus our research on the study of the effects of an external electric field on the optical and electronic properties of electrons confined in two-dimensional QRs. We analyze a 2D GaAs-based single QR whose confining potential model is described via an inverse quadratic function called the Hellmann potential and which allows efficient modeling of interdiffusion effects in the ring material. We also include the effects of a topological defect and report the linear optical absorption coefficient features related to intraband transitions in the presence of a magnetic field applied perpendicular to the structure. The optical properties are analyzed for interlevel transitions from the ground to the first two excited states. In the case of incident resonant radiation, we will study the effects of left- and right-hand circular polarization, which are combinations of mutually perpendicular linear polarizations. Calculations are made in the effective mass and parabolic conduction band approximations. SOI effects are introduced through a $2\times2$ Hamiltonian diagonalized by FEM using the licensed Comsol-Multiphysics 5.4 software. To implement the FEM calculations, we have used meshes with a spatial refinement that allows accounting for the divergences at the system's center, typical of the confinement potential we have chosen. In our opinion, the study is novel, and we consider that it significantly contributes to the knowledge of magnetic properties in 2-dimensional order systems considering SOI effects. The technique for the diagonalization of the $n\times n$ Hamiltonians and the type of problem without symmetries that we present here can be extended to the study of QRs of variable height by implementing the adiabatic approximation. In the same way, this research can be extended to the study of the effects of non-resonant high-intensity incident radiation with linear or circular polarization.

Regarding the research question, at the end of the Introduction section we have added the following text:

The main question of this article and that we answer through this research is: how an in-plane applied electric field and a topological defect can modify the magneto-optical properties relative to states of an electron confined in a quantum ring with Hellmann-like potential using circularly polarized resonant incident radiation?

In the original version of the manuscript, the conclusions did not really reflect the novelties and contributions of our investigation. We have completely rewritten the Conclusions section.

We hope that our comments and answers have been satisfactory and that they allow the Referee to consider that our article has reached an adequate form and that it deserves to be published in the Nanomaterials journal.

Reviewer 3 Report

This work does not so seem to introduce a breakthrough, though, it is interesting to read and can be of some importance for the development of the semiconductor quantum rings.

Although the manuscript is clearly written, English is not so good as it should be. This MS can be recommended for publishing in Nanomaterials, should the authors clarify the following issues:

1.      In the theoretical part, it is relevant to recall what do Rashba and Dresselhaus SOI terms stand for from the physical point of view.  Maybe, it is better to move the paragraph concerning the Rashba and Dresselhaus SOI couplings from the introductory section to the theoretical part.

2.      There is no explanation for the Rashba (α) and Dresselhaus (β) coupling constants. What do these quantifies represent in Eqs. (2) & (3)?

3.      Eqs. (7) & (8) are not correct (there is a mistake in the third term on the left in H11 and H22), see the file attached. If this is not a mere misprint, the authors must have solved their problem using an incorrect Hamiltonian (Eq. (11)).

4. In linear absorption coefficient (Eq. (12)) there is no explanation for the relaxation rate Γ operator.

Author Response

Referee 3

SEE THE PDF FOR DETAILS ON MATHEMATICS FORMULAS

The Referee:

This work does not so seem to introduce a breakthrough, though, it is interesting to read and can be of some importance for the development of the semiconductor quantum rings.

Although the manuscript is clearly written, English is not so good as it should be. This MS can be recommended for publishing in Nanomaterials, should the authors clarify the following issues:

Our reply:

We want to thank the Referee for his comments, which we consider extremely important.

We have presented in the Introduction section some of the details that have motivated this research. We have clearly and forcefully established the research question, which we have tried to answer in the broadest and most precise way throughout our article. Considering the breaking of the axial symmetry of the ring system given the presence of the in-plane electric field and the topological defect, we have presented in the sixth paragraph of the Introduction section the reason that justifies the inclusion in our study of the Rashba and Dresselhaus interaction effects on the Hamiltonian for the electron confined in the quantum ring under applied magnetic field effects.

Regarding the Rahba and Dreselhaus interactions we have added the following paragraph with its corresponding Reference:

In low-dimensional semiconductor systems, the SOI may contribute through two distinct mechanisms. The first one is the Rashba SOI coupling \cite{Vaseghi,Ali}, which arises from the asymmetry of the structure and describes the spin-orbit interaction in low-dimensional systems. In the particular case of heterostructures (semiconductor materials and interfaces between materials), it depends on the shape of the confinement potential.

 The second one is the Dresselhaus coupling \cite{Bejan}, which arises from the lattice inversion asymmetry. In particular, the Dresselhaus coupling describes the spin-orbit interaction caused by the material's crystal structure asymmetry. It is significant for semiconductor materials, where the spin-orbit interaction is weaker than in metals. The Dresselhaus Hamiltonian can be written as a function of the orbital angular momentum and spin operators and has the form of a square matrix. It is a function of the direction of the external magnetic field and the material's properties, such as the effective mass of the electrons and the lattice constant. The SOI may be important for understanding electrical conductivity in carbon nanotubes and quantum dots, where the SOI may cause anisotropy in conductivity, which may have implications for creating electronic devices. Similarly, the SOI also contributes to understanding the physics of spin systems, such as topological spin systems and spin qubits in quantum computing. The SOI can be used to control spin orientation, which is essential for creating spin-robust qubits. All this finds its application in the design of new types of spin-based devices for spintronics. For multiple applications of the SOI, see, for example, Shakouri \textit{et al.} \cite{Shakouri} and references therein.

Regarding the motivation for our article and the news that is introduced, we have written the following paragraphs:

The topic of SOI effects has many open questions that merit further investigation. To a large extent, the investigations reported in the available literature have focused on systems with abrupt potential barriers and quasi-two-dimensional heterostructures with constant heights. In this sense, the effects of the height of the heterostructures have been fixed as a parameter that modifies the coupling constant of the Dresselhaus term \cite{Shakouri}. In large part of QR research, the authors have considered external magnetic fields applied perpendicular to the structure, which preserves the axial symmetry in circular rings. In this way, the obtained wave functions are used efficiently in diagonalization problems of the $n\times n$ Hamiltonians, including SOI effects. The presence of in-plane electric fields breaks the axial symmetry of ring systems. The same situation is obtained by introducing topological defects in the geometry of the structures. The direct consequence of these last two effects translate into the enrichment of the optical transitions allowed between, for example, the system's ground state and the first excited states. Possible applications of such effects include resonant radiation polarization detectors that can be tuned or adjusted through external electric fields. Finally, semiconductor heterostructures present interdiffusion problems at the interfaces of the materials that compose them. So, abrupt potential barriers are still a strong approximation in modeling physical systems.

Considering the open questions raised in the previous paragraph, we focus our research on the study of the effects of an external electric field on the optical and electronic properties of electrons confined in two-dimensional QRs. We analyze a 2D GaAs-based single QR whose confining potential model is described via an inverse quadratic function called the Hellmann potential and which allows efficient modeling of interdiffusion effects in the ring material. We also include the effects of a topological defect and report the linear optical absorption coefficient features related to intraband transitions in the presence of a magnetic field applied perpendicular to the structure. The optical properties are analyzed for interlevel transitions from the ground to the first two excited states. In the case of incident resonant radiation, we will study the effects of left- and right-hand circular polarization, which are combinations of mutually perpendicular linear polarizations. Calculations are made in the effective mass and parabolic conduction band approximations. SOI effects are introduced through a $2\times2$ Hamiltonian diagonalized by FEM using the licensed Comsol-Multiphysics 5.4 software. To implement the FEM calculations, we have used meshes with a spatial refinement that allows accounting for the divergences at the system's center, typical of the confinement potential we have chosen. In our opinion, the study is novel, and we consider that it significantly contributes to the knowledge of magnetic properties in 2-dimensional order systems considering SOI effects. The technique for the diagonalization of the $n\times n$ Hamiltonians and the type of problem without symmetries that we present here can be extended to the study of QRs of variable height by implementing the adiabatic approximation. In the same way, this research can be extended to the study of the effects of non-resonant high-intensity incident radiation with linear or circular polarization.

Regarding the research question, at the end of the Introduction section we have added the following text:

The main question of this article and that we answer through this research is: how an in-plane applied electric field and a topological defect can modify the magneto-optical properties relative to states of an electron confined in a quantum ring with Hellmann-like potential using circularly polarized resonant incident radiation?

After a carefully revisión, we consider that the Conclusions section did not really reflect the novelties and contributions of our investigation. We have completely rewritten the Conclusions section.

We have carefully checked the wording and grammar of our article. We believe that in its revised form it is free from many of the errors that have been typical of the way we traditionally express ourselves.

The Referee:

  1. In the theoretical part, it is relevant to recall what do Rashba and Dresselhaus SOI terms stand for from the physical point of view. Maybe, it is better to move the paragraph concerning the Rashba and Dresselhaus SOI couplings from the introductory section to the theoretical part.

Our reply:

We thank the Referee for his very accurate comment.

In this investigation we have decided to include the Rashba and Dresselhaus terms in the Hamiltonian given the presence of the magnetic field, perpendicular to the quantum ring, and the presence of the in-plane electric field as well as the topological defect, which break the axial symmetry of the problem. In the introduction section we have placed a paragraph where we establish the physics associated with the terms Rashba and Dresselhaus couplings. We ask the Referee to allow us to leave that paragraph in the Introduction section as we want the theoretical model section to be as technical as possible. In the introduction section we have also added a paragraph where we justify the introduction of the Rashba and Dresselhaus effects in our study. We refer the Referee to our answers and comments that we have expressed in the previous item of this letter.

The Referee:

  1. There is no explanation for the Rashba (α) and Dresselhaus (β) coupling constants. What do these quantifies represent in Eqs. (2) & (3)?

Our reply:

We want to thank the Referee for his/her question which has stimulated us to clarify the physical meaning of the constants that appear in equations (2) and (3). This has definitely contributed to a better understanding of our article. After the Eq. (3), we have added the following paragraph with its respective references:

The $\alpha$ constant measures the intensity of the spin-orbit interaction in a material which depends on the material's properties and can vary widely between different materials and crystal structures. The $\beta$ constant describes the spin-orbit interaction caused by the lack of symmetry in the material's crystalline structure. The $\alpha$ and $\beta$ constants can be experimentally determined by different techniques, such as photoemission spectroscopy, scanning tunneling spectroscopy, and the measurement of magnetic anisotropy in low-dimensional systems. Remarkably, the $\alpha$ constant is stronger in materials with an asymmetric crystal structure and low symmetry, such as materials with surfaces and interfaces. The determination of the Rashba and Dresselhaus coupling constants using the conductance of a ballistic nanowire has been reported by M. R. Sakr \cite{Sakr}. The determination of the Rashba spin-orbit coupling strength in InSb nanowire quantum dots under the influence of temperature and nuclear environment has been reported by M. Milivojevic \cite{Milivojevic}. The study has been done via measuring the magnetic susceptibility of the two-electron system in a double quantum dot. Meier \textit{et al.} reported the measurement of Rashba and Dresselhaus spin–orbit magnetic fields \cite{Meier}. They apply their method to GaAs/InGaAs quantum-well electrons, but it should be universally useful to characterize spin–orbit interactions in semiconductors, and therefore could facilitate the design of spintronic devices.

In the first paragraph of the Results and Discusión section we added the following concerning the Dresselhaus and Rashba paramaters:

For the $\alpha$, $\beta$, and $g$- parameters we choose $\beta=10.8$\,meV\,nm, $\alpha=\beta/2$, $g=-2.15$, $\sigma=3.0\times10^{22}$\,m$^{-3}$ \cite{Shakouri}. The $\beta=10.8$\,meV\,nm value is obtained from the bulk Dresselhaus constant ($\beta_b$) as $\beta = \pi^2\,\beta_b/d$, where $d = 5$\,nm is the typical height of the ring structure in the growth direction and $\beta_b = 27.5$ meV\,nm$^3$ for GaAs.

The Referee:

  1. Eqs. (7) & (8) are not correct (there is a mistake in the third term on the left in H11 and H22), see the file attached. If this is not a mere misprint, the authors must have solved their problem using an incorrect Hamiltonian (Eq. (11)).

Our reply:

We want to thank the Referee for his/her comment. We also thank the Referee for the document that he/she has shared with us about how the Hamiltonian expands for an electron confined in a quantum ring in the presence of a magnetic field, perpendicular to the plane of the ring.

We have carefully checked the two diagonal terms of the Hamiltonian and found that they are correct.

In our study, we have made use of the Coulomb gauge (). Additionally, we use the fact that

Considering the Coulomb Gauge, then, results:

In that sense, then, . This solves the problem of a factor of two in the cross term associated with the two operators that enter into the kinetic energy when considering the magnetic field.

In the revised version of the manuscript, we have added the following comment after Eq. (6):

Note that in Eqs. (7) and (8) we have used the fact that for the expansion of the kinetic energy term we are considering the Coulomb gauge ($\nabla\cdot\hat{A}=0$) together with the fact that $\left(\hat{P}+q\,\hat{A}\right)^2\,\psi=\left(\hat{-i\,\hbar\nabla}+q\,\hat{A}\right)^2\,\psi=-\hbar^2\nabla^2\,\psi+q^2\,A^2\,\psi-i\,\hbar\nabla\cdot\hat{A}\,\psi-i\,\hbar\,\hat{A}\cdot\nabla\,\psi$. Additionally, $\nabla\cdot\hat{A}\,\psi=\nabla\cdot(\hat{A}\,\psi)=\hat{A}\cdot\nabla(\psi)+(\nabla\cdot\hat{A})\,\psi=\hat{A}\cdot\nabla(\psi)$. See, for instance, Duque \textit{et al}. \cite{Duque2012} and Dahiya \textit{et al.} \cite{Dahiya}.

The Referee:

  1. In linear absorption coefficient (Eq. (12)) there is no explanation for the relaxation rate Γ operator.

Our reply:

After Eq. (13) we included the following text with its corresponding reference:

In Eq. (12), $\Gamma$ is the phenomenological operator responsible for the dampling due to the electron-phonon interaction, collisions among electrons, etc. In this work, we asume that $\Gamma$ is a diagonal matrix and its element $\gamma_{1j}$ ($j=2,3$) is the inverse of the relaxation time for the state $\psi_j$ \cite{Bloembergen}).

Finally, we hope that our comments and answers have been satisfactory and that they allow the Referee to consider that our article has reached an adequate form and that it deserves to be published in the Nanomaterials journal.

Reviewer 4 Report

This paper focuses on higliting the Spin-Orbit Interaction (SOI) phenomenon in low-dimensional systems and the authors claim that this finds its application in the design of new types of spin-based devices for spintronics showing a remarkable advantages such as high speed, lower energy consumption, and great functionalities.

The contribution discusses about the influence of applying an external electric field to a 2D GaAs-based single Qauntum Ring (QR) whose confining potential model is described via an inverse quadratic function (Hellmann potential). This ersearch adds a topological defect in the quantum ring, whose size is a key factor in introducing modifications in the electronic states. The Hellmann potential was used to model the environment in which an electron is confined in GaAs QR. Parabolic conduction band theory is used, which models lattice effects inside the effective mass approximation.

The results concern the features of linear optical absorption coefficient related to intraband transitions. The geometry was built in the Comsol-Multiphysics 5.4 software that employs the numerical FEM to solve differential equations.

The research plan is clear and the results are clearly presented. Possible applications are discussed.

Author Response

Referee 4

The Referee:

This paper focuses on higliting the Spin-Orbit Interaction (SOI) phenomenon in low-dimensional systems and the authors claim that this finds its application in the design of new types of spin-based devices for spintronics showing a remarkable advantages such as high speed, lower energy consumption, and great functionalities.

The contribution discusses about the influence of applying an external electric field to a 2D GaAs-based single Qauntum Ring (QR) whose confining potential model is described via an inverse quadratic function (Hellmann potential). This rersearch adds a topological defect in the quantum ring, whose size is a key factor in introducing modifications in the electronic states. The Hellmann potential was used to model the environment in which an electron is confined in GaAs QR. Parabolic conduction band theory is used, which models lattice effects inside the effective mass approximation.

The results concern the features of linear optical absorption coefficient related to intraband transitions. The geometry was built in the Comsol-Multiphysics 5.4 software that employs the numerical FEM to solve differential equations.

The research plan is clear and the results are clearly presented. Possible applications are discussed.

Our reply:

We would like to thanks to theReferee for his/her positive words about our work.

Round 2

Reviewer 2 Report

The Authors responded to all my comment in a sufficient manner. Therefore I recommend presented manuscript for publication.